# GroupReduce: Block-Wise Low-Rank Approximation for Neural Language Model Shrinking

**Patrick H. Chen**[*]
UCLA
Los Angeles, CA
patrickchen@g.ucla.edu

**Si Si**
Google Research
Mountain View, CA
sisidaisy@google.com

**Yang Li**
Google Research
Mountain View, CA
liyang@google.com

**Ciprian Chelba**
Google Research
Mountain View, CA
ciprianchelba@google.com

**Cho-Jui Hsieh**
UCLA
Los Angeles, CA
chohsieh@cs.ucla.edu

## Abstract

Model compression is essential for serving large deep neural nets on devices with limited resources or applications that require real-time responses. As a case study, a neural language model usually consists of one or more recurrent layers sandwiched between an embedding layer used for representing input tokens and a softmax layer for generating output tokens. For problems with a very large vocabulary size, the embedding and the softmax matrices can account for more than half of the model size. For instance, the bigLSTM model achieves great performance on the One-Billion-Word (OBW) dataset with around 800k vocabulary, and its word embedding and softmax matrices use more than 6GBytes space, and are responsible for over 90% of the model parameters. In this paper, we propose GroupReduce, a novel compression method for neural language models, based on vocabulary-partition (block) based low-rank matrix approximation and the inherent frequency distribution of tokens (the power-law distribution of words). The experimental results show our method can significantly outperform traditional compression methods such as low-rank approximation and pruning. On the OBW dataset, our method achieved 6.6 times compression rate for the embedding and softmax matrices, and when combined with quantization, our method can achieve 26 times compression rate, which translates to a factor of 12.8 times compression for the entire model with very little degradation in perplexity.

## 1 Introduction

Deep neural nets with a large number of parameters have a great capacity for modeling complex problems. However, the large size of these models is a major obstacle for serving them on-device where computational resources are limited. As such, compressing deep neural nets has become a crucial problem that draws an increasing amount of interest from the research community. Given a large neural net, the goal of compression is to build a light-weight approximation of the original model, which can offer a much smaller model size while maintaining the same (or similar) prediction accuracy.

In this paper, we focus on compressing neural language models, which have been successfully applied in a range of important NLP tasks including language modeling (e.g., next word prediction)

---

[*]Work is done when interning at Google.

and machine translation. A neural language model often consists of three major components: one or more recurrent layers (often using LSTM), an embedding layer for representing input tokens, and a softmax layer for generating output tokens. The dimension of recurrent layers (e.g., LSTM), which corresponds to the hidden state, is typically small and independent of the vocabulary size of input/output tokens. In contrast, the dimension of the embedding and the softmax layers grow with the vocabulary size, which can easily be at the scale of hundreds of thousands. As a result, the parameter matrices of the embedding and softmax layers are often responsible for the major memory consumption of a neural language model. For example, DE-EN Neural Machine Translation task has roughly a vocabulary size around 30k and around 80% of the memory is used to store embedding and softmax matrices. Furthermore, the One Billion Word language modeling task has a vocabulary size around 800k, and more than 90% of the memory footprint is due to storing the embedding and softmax matrices. Therefore, to reduce the size of a neural language model, it is highly valuable to compress these layers, which is the focus of our paper.

There have been extensive studies for compressing fully connected and convolutional networks [20, 5, 7, 6, 25, 27, 9]. The mainstream algorithms from these work such as low-rank approximation, quantization, and pruning can also be directly applied to compress the embedding and softmax matrices. However, it has been reported in previous papers that these algorithms, though efficient for CNN compression, are not able to achieve a good compression rate for word embedding matrices. For instance, [9] proposed a very successful quantization method for CNNs, but for language models the compression rate is less than 3 times.

One important aspect that has not been well explored in the literature is that the embedding matrix has several specific properties that do not exist in a general weight matrix of CNNs. Each column of the input embedding and softmax matrix represents a token, which implies that on a given training or test set the parameters in that column are used with a frequency which obeys Zipf's law distribution.

By exploiting these structures, we propose GroupReduce, a novel method for compressing the embedding and softmax matrices using block-wise, weighted low-rank approximation. Our method starts by grouping words into blocks based on their frequencies, and then refines the clustering iteratively by constructing weighted low-rank approximation for each block. This allows word vectors to be projected into a better subspace during compression. Our experiments show that GroupReduce is more effective than standard low-rank approximation methods for compressing these layers. It is easy-to-implement and can handle very large embedding and softmax matrices.

Our method achieves good performance on compressing a range of benchmark models for language modeling and neural machine translation tasks, and outperforms previous methods. For example, on DE-EN NMT task, Our method achieves 10 times compression rate on the embedding and softmax matrices without much degradation of performance. Results can be further improved to 24 times compression rate when combined with quantization scheme. On One Billion Word dataset, our method achieves 6.6 times compression rate on the embedding and softmax matrices that are originally more than 6GB. When combined with quantization scheme, our method achieves more than 26 times compression rate while maintaining similar perplexity.

## 2 Related Work

### 2.1 Model Compression for CNN

**Low-rank matrix/tensor factorization.** To compress a deep net, a natural direction is to approximate each of its weight matrices, $W$, by a low-rank approximation of the matrix using SVD. Based on this idea, [20] compressed the fully connected layers in neural nets. For convolution layers, the kernels can be viewed as 3D tensors. Thus, [10, 5] applied higher-order tensor decomposition to compress CNN. In the same vein, [8] developed another structural approximation. [12] proposed an algorithm to select rank for each layer. More recently, [27] reconstructed the weight matrices by using sparse plus low-rank approximation.

**Pruning.** Algorithms have been proposed to remove unimportant weights in deep neural nets. In order to do this, one needs to define the importance of each weight. For example, [15] showed that the importance can be estimated by using the Hessian of loss function. [7] considered adding $\ell_1$ or $\ell_2$ regularization and applied iterative thresholding approaches to achieve very good compression rates. Later on, [6] demonstrated that CNNs can be compressed by combining pruning, weight sharing and quantization.

**Quantization.** Storing parameters using lower precision representations has been used for model compression. Recently, [9] showed that a simple uniform quantization scheme can effectively reduce both the model size and the prediction time of a deep neural net. [16] showed that non-uniform quantization can further improve the performance. Recently, several advanced quantization techniques have been proposed for CNN compression [26, 4].

## 2.2 Model Compression for RNN/LSTM

Although model compression has been studied extensively for CNN models, less works have focused on the compression for recurrent neural nets (RNNs), another widely-used category of deep models in NLP applications. Since RNN involves a collection of fully connected layers, many of the aforementioned approaches can be naturally applied. For example, [9] applied their quantization and retraining procedure to compress a LSTM (a popular type of RNN) language model on Penn Tree Bank (PTB) dataset. [24] applied a matrix/tensor factorization approach to compress the transition matrix of LSTM and GRU, and tested their algorithm on image and music classification problems (which does not need word embedding matrices). [19, 17] proposed pruning algorithms for LSTM models compression.

Among the previous work, we found only [9, 17] tried to compress the word embedding matrix in NLP applications. [9] showed that the quantization-plus-retraining approach can only achieve less than 3 times compression rate on PTB data with no performance loss. [17] showed that for word-level LSTM models, the pruning approach can only achieve $87\%$ sparsity with more than $5\%$ performance loss. This means roughly $26\%$ parameters over the original model since this approach also needs to store the index for non-zero locations. Very recently, [14] compressed the word embeddings computed by the word2vec algorithm and applied to similarity/analogy task and Question Answering. [21] applied compositional coding to compress the input embedding matrix of LSTM, but it is challenging to compress the softmax (output) layer matrix using the same algorithm. As a result, the overall compressed model from this approach is still large. One main issue of the approach is that multiple words share the same coding, which makes these words indistinguishable in the output layer during inference.

These previous results indicate that compressing embedding matrices in natural language tasks is a difficult problem—it is extremely challenging to achieve 4 times compression rate without sacrificing performance. In this paper, we will show that instead of only treating the embedding or the softmax parameters as a pure matrix, by exploiting the inherent structure of natural languages, GroupReduce algorithm could achieve much better compression rates.

## 3 Proposed Algorithms

We now introduce a novel algorithm for compressing both the embedding and the softmax layer, two major components in a neural language model as discussed earlier. Assume the word embedding matrix has size $N$-by-$D$, where $N$ is the vocabulary size and $D$ is the embedding dimension. We will use $A \in \mathcal{R}^{N \times D}$ to denote the embedding matrix (either input or softmax layer), and each row of $A$ corresponds to the embedding vector of a word, i.e., the vector representation of the word.

Our goal is to compress the embedding matrix $A$ so that it uses less memory while achieving similar prediction performance. For a typical language model, especially the one with a large vocabulary size, the large memory size of the model is mostly due to the need to store the input and output word embedding matrices. In Table 1, we show an anatomy of memory consumption for several classic models trained on the publicly available datasets. We can see that for three out of four setups, embedding matrices contribute more than 75% of the overall memory usage. For example, in bigLSTM model that achieved start-of-the-art performance on OBW, more than 90% of memory is used to store two (input and output) word-embedding matrices. Thus, for deep neural net models alike, the main challenge to serve them on-device is to store tremendous memory usage of word embedding matrices. As such, it is highly valuable to compress these word embedding matrices.

Given a word embedding matrix $A$, a standard way to compress $A$ while preserving the information is to perform low-rank approximation over $A$. A low-rank approximation can be acquired by using singular value decomposition (SVD), which achieves the best rank-$k$ approximation:

$$A \approx USV^T, \tag{1}$$

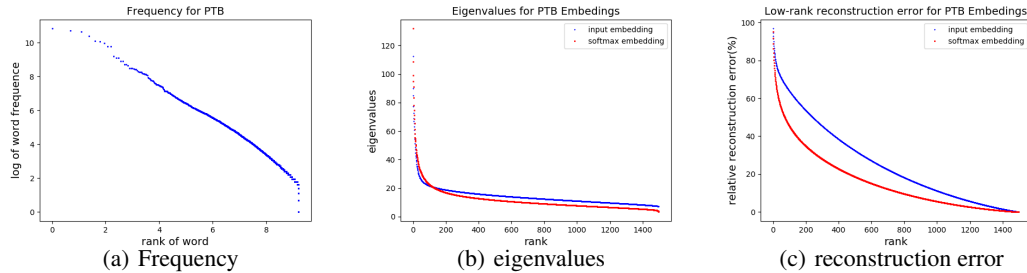

|          | (a) Frequency | (b) eigenvalues | (c) reconstruction error |

Figure 1: Illustration on Penn Treebank (PTB) dataset with the vocabulary size to be 10k and the model's embedding dimension to be 1500. (a): log of word frequency vs rank of the word. One word' rank is defined as the log of number of words that occurs less than it. We can clearly observe the power law distribution of the word frequency; (b) x-axis shows the rank of approximatiion, and y-axis shows the eigenvalues. Here eigenvalues for two embedding matrices are from the input embedding layer and softmax layer; we can see the eigenvalues are very large. (c) low-rank reconstruction error based on singular value decomposition for the two embedding matrices. This in other way shows that the vanilla SVD may not work well for the embedding matrix.

Table 1: The size of each layer in the model. The number in parenthesis shows the ratio respective to the entire model size.

| Models | vocabulary size | dimension | model size | embedding layer(s) | softmax layer | LSTM cell |
|---|---|---|---|---|---|---|
| PTB-Small | 10k | 200 | 17.7MB | 7.6MB(42.9%) | 7.6MB(42.9%) | 2.5MB(14.2%) |
| PTB-Large | 10k | 1500 | 251MB | 57MB(22.7%) | 57MB(22.7%) | 137MB(54.6%) |
| NMT: DE-EN | 30k | 500 | 195 MB | 115 MB (59.0%) | 47MB(24.1%) | 33MB(16.9%) |
| OBW-BigLSTM | 793k | 1024 | 6.8GB | 3.1GB (45.6%) | 3.1GB(45.6%) | 0.6GB(8.8%) |

where $U \in \mathcal{R}^{N \times k}, V \in \mathcal{R}^{D \times k}$ where $k < \min(D, N)$ is the target rank, and $S$ is a diagonal matrix of singular values. After the rank-$k$ low-rank approximation, the memory footprint for $A$ reduces from $O(ND)$ to $O(Nk + Dk)$.

There are two issues for using vanilla SVD to compress an embedding matrix. First, the rank of the SVD is not necessarily low for an embedding matrix. For example, Figure 1(b) shows that all the eigenvalues of the PTB word embedding matrices are quite large, which leads to poor reconstruction error of low-rank approximation in Figure 1(c). Second, the SVD approach considers $A$ as a regular matrix, but in fact each row of $A$ corresponds to the embedding of a word, which implies additional structure that we can further exploit under the language model case.

### 3.1 The Word Frequency Matters

One important statistical property of natural languages is that the distribution of word frequencies can be approximated by a power law. That means a small fraction of words occur many times, while many words only appear few times. Figure 1(a) shows the power-law distribution of word frequency in the PTB datasets.

In the previous compression methods, none of them takes the word frequency into consideration when approximating the embedding matrix. Intuitively, to construct a good compressed model with low-rank approximation under the limited memory budget, it is important to enforce more frequent words to have better approximation. In this paper, we considered two strategies to exploit the frequency information in low-rank approximation: weighted low-rank approximation and block low-rank approximation.

### 3.2 Improved Low-rank Approximation by Exploiting Frequency

**Weighted low-rank approximation.** Firstly, we introduce a weighted low-rank approximation to compress the embedding matrix $A$. This will be used to replace original SVD and serves as the basic building block of our proposed algorithm. The main idea is to assign a different weight for each word's approximation and penalize more for the higher frequency words when constructing low-rank

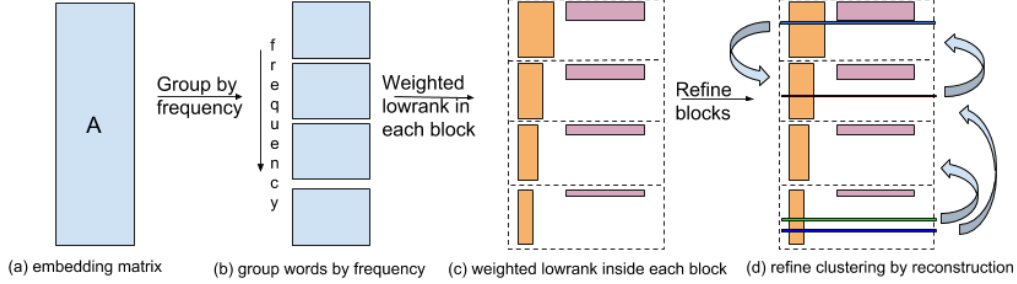

| (a) embedding matrix | (b) group words by frequency | (c) weighted lowrank inside each block | (d) refine clustering by reconstruction |

Figure 2: Illustration of our method. Given an embedding matrix A in (a), we first group the words by their frequency (step (b)), and then perform weighted-SVD inside each group as shown in Eq.2(step (c)). Finally we refine the clustering by considering the low-rank reconstruction error of words as in Eq.5(step (d)).

approximation. Mathematically, for the $i$-th word's frequency to be $q_i$, we want to approximate the embedding $A$ by minimizing

$$\min_{U \in R^{N \times k}, V \in R^{D \times k}} \sum_{i=1}^{N} \sum_{j=1}^{D} q_i (A_{ij} - U_i V_j^T)^2 \tag{2}$$

where $k$ is the reduced rank; $A_{ij}$ is $i$-th word's $j$-th feature; $U \in R^{N \times k}, V \in R^{D \times k}$; $U_i$ and $V_j$ are $i$-th and $j$-th row of $U$ and $V$ respectively. Note that here we do not require $U, V$ to be orthonormal.

Although it is known that weighted SVD with element-wise weights does not have a closed-form solution [23], in our case elements in the same row of $A$ are associated with the same weights, which leads to a simple solution. Define $Q = \text{diag}(\sqrt{q_1}, \dots, \sqrt{q_N})$, then the optimization problem of (2) is equivalent to

$$\min_{U \in R^{N \times k}, V \in R^{D \times k}} \|QA - QUV^T\|_F^2. \tag{3}$$

Therefore, assume all the $q_i$ are nonzeros, we can solve (2) by conducting low-rank approximation of $QA$. Assume $[\bar{U}, \bar{S}, \bar{V}] = \text{svd}(QA)$, then $(U^*, V^*) = (Q^{-1}\bar{U}\bar{S}, \bar{V})$ will be a solution of (2). Therefore solving Eq.(2) is easy and the solution can be immediately computed from SVD of $QA$.

**Block low-rank approximation.** As can be seen from Figure 1(b), the embedding matrix is in general not low-rank. Instead of constructing one low-rank approximation for the entire matrix, we can consider block-wise low-rank approximation–each block has its own approximation to achieve better compression. A similar strategy has been exploited in [22] for kernel approximation (symmetric PSD matrix). Mathematically, suppose we partition the words into $c$ disjoint blocks $\mathcal{V}_1, \dots, \mathcal{V}_c$, and each $\mathcal{V}_p$ contains a set of words. For each block $\mathcal{V}_p$ and its corresponding words' embedding $A_{\mathcal{V}_p}$ in $A$, we can generate a low-rank approximation with rank $k_p$ as $A_{\mathcal{V}_p} \approx U^p (V^p)^T$ for $A_{\mathcal{V}_p}$. Then block low-rank approximation for $A$ is represented as:

$$A = [A_{\mathcal{V}_1}, A_{\mathcal{V}_2}, \cdots, A_{\mathcal{V}_c}] \approx [U^1(V^1)^T, U^2(V^2)^T, \cdots, U^c(V^c)^T]. \tag{4}$$

The challenges for Eq (4) is on how to construct the clustering structure. Intuitively, we want similar frequency words to be grouped in the same block, so we can assign different ranks for different blocks based on their average frequency. For higher frequency words' clusters, we can provide more ranks/budget for better approximation. Meanwhile, we want to make sure the approximation error to be small for words under the same memory budget. Therefore, in this paper we consider two factors, word frequency and reconstruction performance, when constructing the partition. Next, we will explain how to construct the partition.

**Block weighted low-rank approximation.** To take both matrix approximation as well as frequency information into account when forming the block structure in Eq (4), we propose to refine the blocks after initializing the blocks from frequency grouping to achieve lower reconstruction error. In the refinement stage, we move the words around by simultaneously learning a clustering structure as well as low-rank approximation inside each cluster for the word embedding matrix.

Table 2: PTB-small with 5 blocks and 5 times compression rate. We add the proposed strategies one-by-one to see the effectiveness of each of them using the perplexity as the performance metric. Notice that in practice, when applying GroupReduce, we will keep certain percentage of most frequent words uncompressed. But numbers in this table is obtained without preserving any frequent words.

| vanilla SVD | weighted-SVD | block SVD | block weighted-SVD | block weighted-SVD with dynamic rank | refinement |
|---|---|---|---|---|---|
| 161.44 | 155.10 | 143.88 | 135.19 | 129.63 | 127.26 |

Mathematically, given an embedding matrix $A$, we first initialize the blocks by frequency grouping, and then jointly learn both the clustering $\mathcal{V}_1, \mathcal{V}_2, \cdots, \mathcal{V}_c$ and low-rank embeddings for each block $U^p, V^p$ simultaneously by minimizing the following clustering objective:

$$\min_{\{\mathcal{V}_p\}_{p=1}^c, \{U^p\}_{p=1}^c, \{V^p\}_{p=1}^c} \sum_{p=1}^c \|Q_{\mathcal{V}_p} A_{\mathcal{V}_p} - Q_{\mathcal{V}_p} U^p (V^p)^T\|_F^2, \tag{5}$$

where $Q_{\mathcal{V}_p} = \mathrm{diag}_{j \in \mathcal{V}_p}(\sqrt{q_1}, \ldots, \sqrt{q_j})$. Intuitively, the inner part aims to minimize the weighted low-rank approximation error for one cluster, and outer sum is searching for the partitions so as to minimize the overall reconstruction error.

**Optimization:** Eq.(5) is non-convex. In this paper, we use alternating minimization to minimize the above objective. When fixing the clusters assignment, we use weighted SVD to solve for $U^p$ and $V^p$ for each $A_{\mathcal{V}_p}$. To solve for $U^p$ and $V^p$, as mentioned above in Eq(2), we can perform SVD over $Q_{\mathcal{V}_p} A_{\mathcal{V}_p}$ to obtain the approximation. The time complexity is the same with traditional SVD on $A_{\mathcal{V}_p}$.

To find the clustering structure, we first initialize the clustering assignment by frequency, and then refine the block structure by moving words from one cluster to another cluster if the moves can decrease the reconstruction error Eq (5). To compute the reconstruction error reduction, we will project each $A_i$ into each basis $V^p$ and see how much reconstruction error will improve. So if

$$\|A_i - V^p(V^p)^T A_i\| > \|A_i - V^{\bar{p}}(V^{\bar{p}})^T A_i\|, \tag{6}$$

then we will move $i$-th word $A_i$ from the $p$-th cluster to the $\bar{p}$-th cluster. By this strategy, we will decrease the restructure error.

The overall algorithm, GroupReduce is in Figure (2) illustrates our overall algorithm. First, we group the words into $c$ blocks based on frequency. After that, we perform weighted lowrank approximation Eq (2) for each block, and then solve Eq (5) to iteratively refine the clusters and obtain block-wise approximation based on reconstruction error.

There are some implementation details for Algorithm 1. After initial grouping, we assign different ranks to different blocks based on the average frequency of words inside that cluster—the rank $k_p$ for block $p$ is proportional to the average frequency of words inside that cluster. Suppose the block with smallest frequency is assigned with rank $r$, then the rank of cluster $p$ is $\frac{f_p}{f_c}r$, where $f_c$ is the average frequency for the block with least frequency words. $r$ is related to the budget requirement. This dynamic rank assignment can significantly boost the performance, as it assigns more ranks to high-frequency words and approximates them better.

In Table 2, we compare the effectiveness of different strategies in our algorithm. We test on PTB-Small setting with statistics shown in Table 1. Every method in the table has the same compression rate, and we report perplexity number. We compare using vanilla SVD, weighted SVD, weighted SVD for each block (10 blocks), assigning different ranks for different blocks, and refining the blocks. We can see that all the operations involved can improve the final performance and are necessary for our algorithm. The overall memory usage to represent $A$ after our algorithm is $O(Nk + ckD)$, where $N$ is the vocabulary size; $c$ is the number of clusters; $k$ the average rank of each cluster.

## 4 Experiments

### 4.1 Datasets and Pretrained Models

We evaluate our method (GroupReduce) on two tasks: language modeling (LM) and neural machine translation (NMT). For LM, we evaluate GroupReduce on two datasets: Penn Treebank Bank (PTB) and One-billion-Word Benchmark (OBW). OBW is introduced by [2], and it contains a vocabulary

**Algorithm 1:** GroupReduce: Block-Wise Low-Rank Approximation for Neural Language Model Shrinking

---

**Input**: Embedding matrix $A$; number of clusters $c$; the smallest rank $r$; the maximal number of iterations $t_{max}$; minimal size of the candidate set $m_{min}$;

**Output**: Compact representation $\bar{A}$

1  Initialize clusters of words as $\mathcal{V}_1, \mathcal{V}_2, \cdots, \mathcal{V}_c$ by clustering on the frequency of words;

2  Compute the desired rank for each cluster based on the average frequency for that cluster and $r$;

3  **for** $p = 1, \cdots, c$ **do**

4     Compute the rank-$k_p$ weighted lowrank for each sub-matrix $A_{\mathcal{V}_p}$ as $A_{\mathcal{V}_p} \approx U^p(V^p)^T$;

5  **for** $t = 1, \cdots, t_{max}$ **do**

6     $M = []$;

7     **for** $i = 1, \cdots, N$ **do**

8         Compute the reconstruction error for $i$-th word $A_i$, $e^i = min_{p=1\cdots c}\|A_i - V^p(V^p)^T A_i\|_2^2$ ;

9         Find the cluster with smallest reconstruction error $g_i : min_{p=1\cdots c}e_p^i$;

10        **if** $g_i \neq \pi_i$ ($\pi_i$ *is the original cluster index for i-th word*) **then**

11            put $i$ into the candidate set $M$;

12     Choose the top $m$ words in $M$ that with least reconstruction error;

13     move $m$ words (we choose 10% in the paper) into clusters with smallest reconstruction error;

14     **if** $m < m_{min}$ **then**

15         Stop and output;

16     **for** $p = 1, \cdots, c$ **do**

17         **if** *Cluster $\mathcal{V}_p$ changes* **then**

18            Compute the rank-$k_p$ weighted lowrank from Eq (2) for each sub-matrix $A_{\mathcal{V}_p}$ as $A_{\mathcal{V}_p} \approx U^p(V^p)^T$;

19  Output: $\bar{A} = [U^1(V^1)^T, \cdots, U^c(V^c)^T]$

---

of 793,471 words with the sentences shuffled and the duplicates removed. For NMT, we evaluate our method on the IWSLT 2014 German-to-English translation task [1]. On these three benchmark datasets, we compress four models with the models details shown in Table 1. All four models use a 2-layer LSTM. Two of them (OBW and NMT) are based on exiting model checkpoints and the other two (based on PTB) are trained from scratch due to the lack of publicly released model checkpoint.

We train a 2-layer LSTM-based language model on PTB from scratch with two setups: PTB-Small and PTB-Large. The LSTM hidden state sizes are 200 for PTB-Small and 1500 for PTB-Large, so are their embedding sizes. For OBW, we use the "2-LAYER LSTM-8192-1024" model shown in Table 1 of [11]. For NMT, we use the PyTorch checkpoint provided by OpenNMT [13] to perform German to English translation tasks. We verified that all these four models achieved benchmark performance on the corresponding datasets as reported in the literature. We then apply our method to compress these benchmark models.

For experiments using BLEU scores as performance measure, we report results when the BLEU scores achieved after compression is within 3 percent difference from original score. For experiments using perplexity (PPL) as measure such as PTB dataset, we target 3 percent drop of performance too. For OBW dataset, since it has larger vocaburary size, we report results within 10 percent difference from original PPL. For each method in Table 3, 4 and 5, we tested various parameters and report the smallest model size of the compression fulfilling above criteria. Certainly, the compression rate and corresponding performance will be a spectrum. The more we compress, the larger the performance drop. We plot this trade-off on PTB-Large in the supplementary. Number of clusters will impact the compression rate. In the experiment, we set the number of clusters to be 5 for PTB and IWSLT datasets, and 20 for the OBW dataset. We show the performance of GroupReduce with different numbers of clusters under the PTB-Large setting in the supplementary.

Note that the goal of this work is to compress an existing model to a significantly-reduced size while maintaining accuracy (e.g., perplexity or BLEU scores), rather than attempting to achieve higher

Table 3: Embedding compression results on three datasets comparing our method GroupReduce with Low-rank and Pruning. Compression rate is compared to both input embedding and softmax layer. For example, 10x means approximated embedding uses 10 times smaller memory compared to original input layer and softmax layer.

| Model | Metric | Original | Low-rank | Pruning | GroupReduce |
|---|---|---|---|---|---|
| PTB-Small | Embedding Memory | 1x | 2x | 2x | 4x |
| | PPL(before retrain) | 112.28 | 117.11 | 115.9 | 115.38 |
| | PPL(after retrain) | – | 113.83 | 113.78 | 113.81 |
| PTB-Large | Embedding Memory | 1x | 5x | 3.3x | 8x |
| | PPL(before retrain) | 78.32 | 84.63 | 84.23 | 84.79 |
| | PPL(after retrain) | – | 80.04 | 78.38 | 79.83 |
| OBW-bigLSTM | Embedding Memory | 1x | 2x | 1.14x | 6.6x |
| | PPL(before retrain) | 31.04 | 39.41 | 128.31 | 32.47 |
| | PPL(after retrain) | – | 38.03 | 84.11 | 32.50 |
| NMT: DE-EN | Embedding Memory | 1x | 3.3x | 3.3x | 8x |
| | BLEU(before retrain) | 30.33 | 29.65 | 25.96 | 29.31 |
| | BLEU(after retrain) | – | 29.96 | 29.34 | 29.96 |

accuracy. It is possible that there are models that could achieve higher accuracy, in which case our method can be applied to compress these models as well.

## 4.2 Comparison with Low-Rank and Pruning

We compare GroupReduce with two standard model compression strategies: low-rank approximation and pruning.These two techniques are widely used for language model compression, such as [17, 19, 18] We compress both input embedding and softmax matrices. For the low-rank approximation approach, we perform standard SVD on the embedding and softmax matrices and obtain the low-rank approximation. For pruning, we set the entires whose magnitude is less than a certain threshold to zero. Note that storing the sparse matrix requires to use the Compressed Sparse Row or Compressed Sparse Column format, the memory usage is thus 2 times the number of non-zeros in the matrix after pruning. After approximation, we retrain the rest of parameters by SGD optimizer with initial learning rate 0.1. Whenever, the validation perplexity does not drop down, we decrease the learning rate to an order smaller. As shown in Table 3, GroupReduce can compress both the input embedding and softmax layer 5-10 times without losing much accuracy. In particular, GroupReduce compress 6.6 times on the language model trained on OBW benchmark, which saves more than 5 GB memory.

Notice that GroupReduce achieves good results even before retraining. This is important as retraining might be infeasible or take a long time to converge. We experimented with different learning rates and retrained for 100k steps (about 3 hours), but we observe that all the retraining scheme of OBW-bigLSTM model after approximation do not lead to significant improvement on accuracy. One reason is that to retrain the model, we need to keep the approximated embedding matrices fixed and re-initialize other parameters, and train these parameters from scratch as done in [21]. On OBW-bigLSTM, it will take more than 3 weeks for the retraining process. It is not practical if the goal is to compress model within a short period of time. Therefore, performance before retraining is important and GroupReduce in general obtains good results.

## 4.3 Comparison with Quantization

As noted in the related work, quantization has been shown to be a competent method in model compression [9]. We implement b-bit quantization by equally spacing the range of a matrix into $2^b$ intervals and use one value to represent each interval. For example, 4-bit quantization will transform original matrix into matrix with 16 distinct values.

We need to point out that quantization is not orthogonal to other methods. In fact, GroupReduce can be combined with quantization to achieve a better compression rate. We firstly approximate the embedding or the softmax matrices by GroupReduce to obtain low rank matrices of each block, and then apply 4 or 8 bits quantization on these low rank matrices. After retraining, quantized GroupReduce could achieve at least 26 times compression for both input embedding and softmax matrix in OBW as shown in Table 4. In addition, comparisons to other coding schemes including deep compositional coding [21] and dictionary coding [3] are shown in the supplementary.

Table 4: Embedding compression results on three datasets comparing our method Quantized GroupReduce with traditional Quantization. 10x means approximated embedding uses 10 times smaller memory compared to original input embedding layer and softmax layer.

| Model | Metric | Original | Quantization | Quantized GroupReduce |
|---|---|---|---|---|
| PTB-Small | Embedding Memory | 1x | 6.4x | 16x |
| | PPL(before retrain) | 112.28 | 115.81 | 116.54 |
| | PPL(after retrain) | – | 114.14 | 114.39 |
| PTB-Large | Embedding Memory | 1x | 6.4x | 20x |
| | PPL(before retrain) | 78.32 | 81.69 | 81.53 |
| | PPL(after retrain) | – | 79.22 | 78.61 |
| OBW-bigLSTM | Embedding Memory | 1x | 6.4x | 26x |
| | PPL(before retrain) | 31.04 | 32.63 | 34.43 |
| | PPL(after retrain) | – | 33.86 | 33.60 |
| NMT: DE-EN | Embedding Memory | 1x | 6.4x | 32x |
| | BLEU(before retrain) | 30.33 | 27.41 | 29.33 |
| | BLEU(after retrain) | – | 30.19 | 29.65 |

Table 5: Compression rate of overall model compression using Quantized GroupReduce. Compression rate shown in the column 4-6 is compared to the corresponding part of the model.

| Models | Original PPL/BLEU | PPL/BLEU after approximation | input layer | softmax layer | LSTM cell | Overall Compression |
|---|---|---|---|---|---|---|
| NMT: DE-EN | 30.33(BLEU) | 29.68(BLEU) | 24x (45.9%) | 24x(31.8%) | 4x(22.3%) | 11.3x |
| OBW-BigLSTM | 31.04(PPL) | 33.61(PPL) | 26x (45.6%) | 26x(45.6%) | 2x(8.8%) | 12.8x |

## 4.4 Overall Compression

Results above have shown GroupReduce is an effective compression method when the frequency information is given. We need to point out that part of the model (e.g., LSTM cells) cannot leverage this information as the transition matrices in LSTM cell do not correspond to the representation of a word. We adopt simple quantized low-rank approximation of LSTM to compress this part. Specifically, we first compute SVD of LSTM matrix to obtain 2 times compression, and quantize the entries of low-rank matrices by using only 16 bits. In total the model would be 4 times smaller. However, we found out for OBW-bigLSTM model, LSTM matrix does not have a clear low-rank structure. Even slight compression of LSTM part will cause performance significantly drop. Therefore, we only apply 16-bit quantization on OBW-bigLSTM to have a 2 times compression on LSTM cells. Overall compression rate is shown in Table 5. With the aid of GroupReduce, we can achieve over 10 times compression on both language modeling and neural machine translation task.

## 5 Conclusion

In this paper, we propose a novel compression method for neural language models. Our method leverages the statistical property of words in language to form block-wise low-rank matrix approximations for embedding and softmax layers. Experimental results show that our method can significantly outperform traditional compression methods such as low-rank approximation and pruning. In particular, on the OBW dataset, our method combined with quantization achieves 26 times compression rate for both the embedding and softmax matrices, which saves more than 5GB memory usage. It provides practical benefits when deploying neural language models on memory-constrained devices. For the future work, we will investigate different retrain schemes such as training the block low-rank parameterization of the model end-to-end.

## 6 Acknowledgement

This research is mainly done during Patrick Chen's internship at Google Research. We also acknowledge the support by NSF via IIS-1719097, Intel faculty award, Google Cloud and Nvidia.

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
