[Supplementary Material · NIPS_GroupReduce_Camera_ready_suppl_New_Stye.pdf]

# GroupReduce: Block-Wise Low-Rank Approximation for Neural Language Model Shrinking - Supplementary

**Patrick H. Chen**[*]
UCLA
Los Angeles, CA
patrickchen@g.ucla.edu

**Si Si**
Google Research
Mountain View
sisidaisy@google.com

**Yang Li**
Google Research
Mountain View
liyang@google.com

**Ciprian Chelba**
Google Research
Mountain View
ciprianchelba@google.com

**Cho-Jui Hsieh**
UCLA
Los Angeles, CA
chohsieh@cs.ucla.edu

## 1 Selection of the Number of Clusters

In our method, the number of clusters is a hyperparameter for groupReduce. We experimented with different numbers of clusters on the PTB-Large setup with 6.6 times compression (e.g., using only 15% of the memory compared to the original matrices) of both input embedding and softmax matrix, and the results are shown in Table 3. As we can see from the table, our method is robust to the number of clusters. In the experiments with the PTB and IWSLT dataset, we set the number of clusters to be 5. On the OBW dataset, as the vocabulary size is larger so we set the number of clusters to be 20.

## 2 Comparison with Deep Compositional Coding

Since deep compositional coding [2] can only compress input embedding matrix, to demonstrate the effectiveness of GroupReduce, we compare with GroupReduce on only compressing input embedding matrix. We evaluate results based on NMT:DE-EN and PTB-Large setups. Again, after compressing each model, we retrain the model while keeping the input embedding fixed. We use SGD with learning rate 0.1 as the start, and lower the learning rate an order whenever validation loss stops decreasing. Results are summarized in Table 2. As shown in the table, GroupReduce can compress twice better than deep compositional coding. More importantly, GroupReduce can be applied to both input and softmax embedding which makes overall model not just input embedding smaller as shown in the experiment section.

## 3 Comparison with Dictionary Coding

Dictionary Coding [1] is another method to compress the softmax or embedding layer. The main intuition is that some words in dictionary can be represented by other words, and it assumes the embedding of low frequency words is linear combination of embedding of high frequency words. Our method differs from 1) They partitioned words into only 2 clusters (low and high frequency) while we are flexible about the number of clusters and we perform refinement to adjust the clustering; (2) We perform weighted low-rank, which takes frequency into consideration when constructing the low-rank. Experimentally, for PTB-large setup (choosing 1k common words and learning embedding of uncommon words by solving constrained sparse coding), [1]'s compression rate is 5x with PPL to

---

[*]Work is done when interning at Google.

| (a) Spectrum of compressing both layers | (b) Input or Softmax only |

Figure 1: Illustration on Penn Treebank (PTB) dataset with the vocabulary size to be 10k. (a) Perplexity versus compression rate plot. As we can see that when compressing the model toward less than 5 percent of original model, perplexity goes up quickly. (b) Compression of input layer affects perplexity less than compression of Softmax layer.

be 93.24 before retrain and 81.8 after retrain, while our method could achieve 10x with PPL to be 79.16 after retrain.

## 4 Compression Spectrum

We only report the compression rate and PPL or BLEU score in the tables in the experiment section; however, as we can imagine that different compression rates will lead to different accuracy. Therefore, the perplexity and compression rate function will be a spectrum. We include one plot in 1(a) on PTB-large setup to illustrate this point. Blue bullet point in the figure is the number we reported in the table in the experiment section.

How much we can compress while maintaining accuracy is different for input embedding or softmax layer. We conduct compression with input emebdding or softmax layer only and plot their spectrums in 1(b). As we can see that we can compress the input layer more than the softmax layer.

## 5 Qualitative Results

We select some translated sentences of DE-EN task shown in Table 1 to demonstrate that our algorithm can provide similar translations but with smaller memory usage.

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

Table 1: Qualitative comparison of our method to uncompressed computation. The compressed model used is the same as reported in Table 3 in the main text with reported BLEU 29.96.

| Full-softmax | Our method |
|---|---|
| we have to deal with the <unk> separation of death , and so **it shouldn't surprise us** that we all sing , dance and art . | we have to deal with the <unk> separation of death, and so **we should not surprise** that we all sing , dance and art . |
| all of these people teach us that there are other <unk> , other ways , other ways of **thinking about the earth's impact.** | all of these people teach us that there are other <unk> , other ways , other ways of **engaging on earth .** |
| and that's an idea that when you think about it , you can only **fulfill** one with hope . | and that's an idea that when you think about it , you can only **meet** one with hope . |
| we **followed** the <unk> eight generations , and we found two cases of one natural death , and when we <unk> people with <unk> , they decided that one of the people had become so old that he died because of his age , and we killed him with the **spear anyway** . | we **came back to** <unk> eight generations , and we found two cases of one natural death , and when we <unk> people with <unk> , they decided that one of the people had become so old that he died because of his age , and we killed him with the **rider** . |
| the young <unk> are separated at the age of three and four years from their families , and in a <unk> world of darkness , **in <unk> of** the glacier for 18 years , **at the base of** the glacier for 18 years , selected to mimic the nine months of **pregnancy** that they spent in their mother's lap , **and** they're now spending <unk> **in the lap of great mother .** | the young <unk> are separated at the age of three and four years from their families , and in a <unk> world of darkness , **at the end of** the glacier **at the end of** the glacier for 18 years , **and two times the age of nine years ,** <unk> to mimic the nine months of **pregnant** that they spent in their mother's lap , they're now spending <unk> **in the great mother's lap .** |
| **whenever** we think about **native** and landscape , we're either <unk> , and the old **fragment** of the **fancy** <unk>, which is a <unk> thought , or <unk>, and **say** these people are more connected to nature than we are . | **every time** we think about **<unk>** and landscape , we're either <unk> , and the old **<unk>** of the **male** <unk> , which is a <unk>thought , or <unk> , and **saying** these people are more connected to nature than we do . |

Table 2: Comparison of input embedding compression results on two datasets. Note that the numbers in the table is the compression rate based on only input embedding not overall model size.

| Model | Metric | Original | Deep Compositional Coding | Quantized GroupReduce |
|---|---|---|---|---|
| PTB-Large | Embedding Memory | 1x | 11.8x | 23.6x |
| | PPL(before retrain) | 78.32 | 81.82 | 80.20 |
| | PPL(after retrain) | – | 79.58 | 79.18 |
| NMT: DE-EN | Embedding Memory | 1x | 16.6x | 33.3x |
| | BLEU(before retrain) | 30.33 | 28.85 | 28.89 |
| | BLEU(after retrain) | – | 29.97 | 30.16 |

Table 3: GroupReduce with different number of clusters. Results are evaluated on PTB-Large setup with 6.6 times compression rate on both input embedding and softmax layer.

| Number of Clusters | 5 | 10 | 20 | 30 |
|---|---|---|---|---|
| PPL(before retrain) | 81.79 | 80.52 | 82.88 | 83.1 |
| PPL(after retrain) | 78.44 | 78.5 | 78.52 | 80.1 |