[Reviews · NeurIPS 2018]

Reviewer 1



The paper presented an algorithm to compress the word embedding matrices in a neural language model. The algorithm, albeit quite straightforward, is shown to achieve significant model compression without incurring much performance loss. The paper is largely well written and easy to follow. * From Table 2 it seems the "weighted-lowrank" treatment (and "refinement" as well) isn't that effective when compared to block-wise decomposition (w/ or w/o dynamic rank). Therefore it might be beneficial to conduct experiments with block-wise decomposition **alone** to help clarifying whether this is indeed the case. * The authors might want to reference and compare with the following paper which is essentially solving the same task (and also achieving promising results), albeit using a different technical solution. Chen et al., Compressing Neural Language Models by Sparse Word Representations, ACL 2016

Reviewer 2



The sizes of embedding matrices in NLP tasks have long posed difficult computational problems, either from the inefficiency of operating (softmaxing) over them, or often from the sheer difficulty in storing them. In this paper the authors take on the latter problem, introducing a method of using multiple low-rank approximations to reduce the size of these matrices. They rely on frequency binning — the same observation underlying the hierarchical softmax solution to the former problem — to group words, prioritizing the most frequent words to receive higher rank approximations. This itself leads to significant compression rates with little loss in accuracy, and when further combined with quantization, yields large reduction in memory. Importantly quantization appears to play nicely with their methodology, and the combined seem to provide much smaller models overall, while performing at least as well as naive quantiziation on large data sets. The method is simple but effective, and the paper is clearly written, the problem of obvious importance, so my main critique comes from _how_ the material is presented. (1) I thought some of the comparisons were more arbitrary than I’d like. The size of the parameter reduction vs. performance is, of course, a spectrum. So, for instance, to present a 40x reduction on small LSTM, when the perplexity hit was huge, is not so informative IMO. It’s already a small model — it’s less interesting to see it reduced 40x vs. the larger ones. I’d be more interested to see how much it can be reduced while retaining approximately full performance level. This could be said for many of the presented results — they all seem arbitrarily chosen and while some provide clear take-aways, these are very small glimpses into the larger size-accuracy trade-off at hand. I’m even tempted to say that, despite what appear to be very compelling results in specific cases on an important problem, and a technique with great practical importance, I’d rather see it resubmitted with more attention paid to painting a clear picture of these trade-offs. (2) Some things were over-stated. Biggest offender: "On the OBW dataset, our method achieved 6.6x compression rate for the embedding and softmax matrices, and when combined with quantization, our method can achieve 26x compression rate without losing prediction accuracy.” In this setting perplexity goes from 31.04 to 32.47. I don’t want to claim this is going to make any practical difference, but I don’t understand the need to overstate the effectiveness of this approach when it already does so well. This is easy to fix so please ammend strong but unsupported claims. Question for the authors: If perplexity is all one cares about, and each token is considered uniformly important, than such frequency binning here is a well-motivated decision. Still, I wonder for many domains where many of the most frequent words are not so semantically interesting, if this will have a negative effect on end-task performance. Perhaps something like speech recognition, or other product models, where the most frequent words are seen often by the core model and the language model provides little useful information then about them, but now provides poorer approximations to rarer words the core model is also likely to misidentify. Have the authors looked into a more qualitative error analysis of where the reduced models goes wrong? Or along these lines, is there anything to say about the post-GroupReduce cluster membership? Are there any conceptual trends in how the goal of reducing reconstruction error adjusted the freq-based initial partitionings? Can you comment on the computation time of these experiments? Is there a noticeable performance difference on compressing just the embedding matrix vs. just the output matrix? I’m curious if the compression “gains” come from both about equally, or are coming primarily from one or the other. Misc: "One word’ rank is" --> typo Table 2 needs a label of its metric within the figure "The overall algorithm is in Algorithm 1. Figure 2 illustrates our overall algorithm." —> confusing "due to the lack of publicly released model checkpoint." —> grammar "2-LAYER LSTM-8192-1024” —> this model is not shown as mentioned AVp as AVp ≈ Up(V p)T —> not sure if this is a type or odd phrasing, but could ditch the “as AVp"

Reviewer 3



This paper proposes a method for reducing the number of parameters used by standard language models at test time by exploiting the power law distribution of words in language. The authors are able to reduce the number of parameters by factors of 5-10 (24-40 when combined with other methods), while maintaining roughly the same language model perplexity or machine translation BLEU accuracies. The method proposed in this paper seems intuitive: infrequent words can be compressed using less reconstruction accuracy than frequent words. The authors cluster words by their frequency and compute different SVD decompositions with different ranks, which are selected based on these frequencies. They then reconstruct the original embedding matrices at test time, resulting in a much more memory efficient model. Importantly, this trick applies to both the input and output layer. I like the ideas in this paper, and agree that empirically they work well. However, the ideas are not entirely novel, and the solution is somewhat limited in scope (see below). Comments: 1. The idea to exploit word frequencies in embedding matrix was used in the past by [1]. While they only applied this method to the output layer, and showed smaller memory gains, their method (a) works at training time, and thus is much more useful, and (b) is not limited to improvement in terms of space, but also leads to reduction in both training and test time. In contrast, the proposed method increases training time by computing the SVDs (the authors do not mention by how much, but do say that they can't run it for too long as it would take too much time, line 265). To the best of my understanding, it also increases test time. I would be happy if the authors could in their response explain the runtime effect of the proposed method. 2. There are other ways to significantly reduce the number of parameters: (a) character embeddings is a memory-efficient alternative to using word embeddings (albeit also at the expanse of runtime). As shown by [2] (table 1), character embeddings require substantially less memory, while getting slightly better perplexity. (b) tying the input and output embeddings also reduces the number of parameters significantly [3]. 3. Given the power law distribution of words, I would expect to see the 1-2 small clusters of very high frequency words, followed by a few very large clusters of low frequency words. To the best of my understanding, the authors initialized their model such that all clusters are the same size, and only modified them later in the second step? this looks like a sub-optimal selection. I would have liked to see the final size of the clusters. 4. 237: "We verified that all these four model achieved benchmark performance on the corresponding datasets as reported in the literature": state-of-the-art on PTB is quite lower [4] 5. No mention of hyperparameter values (e.g., r) 6. No summary section 7. Table 1 shows number of parameters in MB and GB. It would be nice to see the actual number of parameters. 8. Why are the numbers in Table 2 inconsistent with Table 4? Also, a baseline (uncompressed model) is missing in Table 2. Typos and such: 96: "Among the previous work, we found only [8, 16] *tried* to compress the word embedding matrix in NLP applications.": *who* tried? 104: "However, as they *explicit* mentioned ...": explicitly 121: "For a typical language model, especially *with one with* a large vocabulary" 213: "This dynamic rank assignment can *significant* boost the performance...": significantly References: [1] "Efficient softmax approximation for GPUs", Grave et al., ICML 2017 [2] "Exploring the limits of language modeling", Jozefowicz et al., 2016 [3] "Using the Output Embedding to Improve Language Models", Press and Wolf, EACL 2017 [4] "Regularizing and optimizing LSTM language models", Merity et al., ICLR 2018 ===== Thank you for your response. It is encouraging to see that the method doesn't lead to significant increase in training time, and even runs faster at test time. The wordpiece results are also reassuring. I've updated my score to 7.